# Are Hair Cortisol Levels of Humans, Cats, and Dogs from the Same Household Correlated?

**DOI:** 10.3390/ani12111472

**Published:** 2022-06-06

**Authors:** Justyna Wojtaś, Aleksandra Garbiec, Mirosław Karpiński, Patrycja Skowronek, Aneta Strachecka

**Affiliations:** 1Department of Animal Ethology and Wildlife Management, University of Life Sciences in Lublin, Akademicka 13, 20-950 Lublin, Poland; justyna.wojtas@up.lublin.pl (J.W.); miroslaw.karpinski@up.lublin.pl (M.K.); 2Department of Invertebrate Ecophysiology and Experimental Biology, University of Life Sciences in Lublin, Akademicka 13, 20-950 Lublin, Poland; patrycja.skowronek@up.lublin.pl (P.S.); aneta.strachecka@up.lublin.pl (A.S.)

**Keywords:** hair cortisol level, human-animal interaction, dog, cat

## Abstract

**Simple Summary:**

Dogs and cats are animals that have been accompanying humans for many years. There is no doubt that they are emotionally connected with people, although each of them in their own way. The study attempts to assess the emotional relationship between humans, dogs, and cats living in one household on the basis of the correlations between the hair cortisol level. The study involved 25 women who had at least one dog and at least one cat at home. Based on the study conducted, no significant correlation was found between the level of cortisol in the hair of the owners and their pets. There were, however, some interesting differences depending on the degree of emotional connection and the frequency of interactions.

**Abstract:**

Human–animal interactions and the emotional relationship of the owner with the pet are the subjects of many scientific studies and the constant interest of not only scientists but also pet owners. The aim of this study was to determine and compare the hair cortisol levels of dogs, cats, and their owners living in the same household. The owners were asked to complete a questionnaire concerning the frequency of their interactions with pets and emotional relationship with each of their cats and each of their dogs. The study involved 25 women who owned at least one dog and at least one cat. In total, 45 dogs and 55 cats from 25 households participated in the study. The average level of hair cortisol of the owners was 4.62 ng/mL, of the dogs 0.26 ng/mL, and in the hair of cats 0.45 ng/mL. There was no significant correlation between the hair cortisol level of the owner and dog or the owner and the cat and between dogs and cats living together. A significant positive correlation was observed between the hair cortisol level in the owner and the pet, for dogs in which the owner performs grooming treatments once a week and for cats which are never kissed. Although our study did not find many significant correlations, studies using other stress markers might have yielded different results.

## 1. Introduction

Human–animal interactions are one of the key issues of interest to psychologists, zoopsychologists, and scientists in the field of behavior and animal welfare, both with respect to companions [1,2,3] and farm animals [4]. The companion animals are now treated as family members. In every family, including interspecies ones, there are a number of relationships, interactions, and conflicts that affect the stress level, and thus the wellbeing of individual members of the interspecies “herd.” The relationship between the dog’s and its handler or owner’s reaction to stress has been examined. Acute stress was assessed by Buttner et al. [5] and Wojtaś et al. [3]. Chronic stress was assessed by Sundman et al. [6]. The behavioral and physiological effects of dog–human interactions were described by Payne et al. [7], Petersson et al. [8], and Willen et al. [9]. The human–cat relationship has been extensively analyzed by Turner [10]. The effects of humans on cats based on oxytocin and cortisol levels in urine were analyzed by Nagasawa et al. [11]. The effect of cats on humans was investigated by Turner et al. [12].

There are at least a few reasons why hair is increasingly used and appreciated as biological material in research in many fields. It is a material of high durability and resistance to external factors. As hair does not appear suddenly but grows over weeks or months, analysis of its composition makes it possible to measure physiological changes over a given time scale. Research shows that the hair growth rate is about 1 cm/month [13,14,15], therefore it is assumed that the presence of a given substance in 1 cm of a hair length corresponds to a period of about a month’s exposure to a given substance [16]. In this way, it is possible to evaluate the organism’s exposure to toxins retrospectively or to detect hormonal changes related to disease or pregnancy [17].

Social interactions and group relationships are important modulators of the activity of the HPA (hypothalamic–pituitary–adrenal) axis [18]. The HPA axis is a major component of the physiological stress response in mammals. Its activity can be analyzed by assessing the level of cortisol. Cortisol, known as the stress hormone, has a wide range of effects on how the body works. It participates in the metabolism of glucose and lipids, and has immunosuppressive and anti-inflammatory effects [19]. It influences protein, calcium, and water–electrolyte metabolism as well as hematopoietic systems [20]. Chronic stress is thought to be accompanied by a hyperactive HPA axis [19]. Measuring hair cortisol is a potential method of measuring the long-term stress response and reflects the average level of circulating cortisol built into growing hair over time [21].

Since 2004, when the first report on the measurement of cortisol in hair appeared [22], the interest in its laboratory potential and possible applications in the assessment of chronic stress levels was gradually increasing [23]. Studies on the HCL of dogs were conducted, among others, by Accorsi et al. [24] and Direksin et al. [25]. The level of HCL of cats has been studied by Accorsi et al. [24] and Franchini et al. [26]. As for other animal species, cortisol has been studied in, among others, sheep wool [27], pig hair [28], dairy cows [29], horses [30], and captive wild animals [31,32].

The aim of this study was to determine the hair cortisol level (HCL) of dogs, cats, and their owners living in the same house. To our knowledge, hormonal relationships in dogs and cats, dogs and their owners, and cats and their owners living all together have not yet been evaluated. We assumed that there would be hormonal correlations between individuals living in one household. Correlations between the levels of stress hormones in humans and their pets have already been observed in our previous studies [3] on acute stress as well as in Sundman’s studies [6] on chronic stress.

## 2. Materials and Methods

All methods used in the study were in line with the Act of 15 January 2015 on the protection of animals used for scientific or educational purposes (Journal of Laws of 2015, item 266) and the Directive of the European Parliament and of the Council of 22 September 2010 on the protection of animals used for scientific purposes (2010/63/EU). The tests performed were non-invasive, which means that, within the meaning of the Directive, they did not cause pain, suffering, distress, or permanent damage to an extent equal to, or more severe than, a needle-stick injury. All owners were over 18 years of age and gave their written consent to participate in the study voluntarily.

Twenty-five women who owned at least one dog and at least one cat were invited to take part in the study. The invitation to participate in the study was made available on the internet, on a social network, in groups associating animal lovers. The condition for participation in the study was to have at least one dog and at least one cat at home. In total, 45 dogs and 55 cats from 25 households participated in the study. The women belonged to two age groups: <25 (*n* = 10) and 25–50 (*n* = 15). More than half (*n* = 14) indicated a village as their place of residence, the others (*n* = 11) indicated the city. The study participants included 15 working women, 6 working students, and 4 non-working students. The group of dogs included 27 females and 18 males. Of all dogs, 31 were castrated. Three dogs were under 2 years of age, 22 dogs were aged 2–6 years, and 20 dogs were 7 years or older. Fifty-five cats took part in the study—31 females and 24 males. There were 50 cats after the castration procedure and only 5 animals were not castrated. The animals were divided into three age groups—less than 3 years old (*n* = 14), 3–10 years old (*n* = 29), and 11 and more years old (*n* = 12). All animals participating in the study, as well as their owners, did not undergo any surgery for at least 6 months preceding the study and did not receive treatment for chronic diseases.

The women were asked to complete a questionnaire that was a shortened modification of the MDORS scale—Monash Dog-Owner Relationship Scale [33] and CORS—Cat-Owner Relationship Scale [34]. The first part of the questions concerned basic information about the owner, cat, and dog, such as gender, age, and place of residence. The second part of the questionnaire concerned the frequency of individual owner interactions with each of her cats and each of her dogs. In the third part of the questionnaire, the owner rated her emotional relationship with each of her cats and each of her dogs. The second and third parts of the questionnaire are provided as Appendix A.

The biological material in the study was hair. The hair was collected in a non-invasive way, by cutting right next to the skin. Hair was collected from the lumbosacral area from dogs and cats and from the occipital area of the owners. For the analysis, the centimeter closest to the scalp end was used. Taking a hair sample is a simple procedure that can be performed after a short briefing and does not require the presence of a professional. The cutting of the hair is minimally invasive and painless [35]. The hair samples were placed in foil bags and stored at room temperature until analysis [23].

The extraction methodology was modified from Koren et al. [36] and Accorsi et al. [24]. Hair was first minced into 1–2 mm length fragments and 20 mg of trimmed hair was put in a glass vial. Three-and-a-half methanol (Sigma-Aldrich, Poznań, Poland) was added, and vials were incubated at 50 °C with gentle shaking for 24 h. After incubation, the supernatant was filtrated to separate the liquid phase and put into disposable glass culture tubes. Following this, this supernatant was evaporated to dryness under an air-stream suction hood at 37 °C. Dry residue was then dissolved into 1 mL of phosphate-buffered saline (PBS) 0.05 M, pH 7.5. Samples were vortexed for one minute followed by another 30 s until they were well mixed. The cortisol levels in the samples were determined with the DRG Salivary Cortisol HS ELISA assay. The procedures followed the manufacturer’s instructions. All samples were measured in triplets. Cortisol concentrations were expressed in ng/mL.

The statistical analysis was performed with the use of the Statistica 13.3 statistical package. The analysis of the correlation between the HCL in the pets and their owners was performed based on Spearman’s rank correlation coefficient, due to the deviations of the cortisol level distribution from the normal distribution. The compliance of the distributions with the normal distribution was assessed with the Shapiro–Wilk test. The analysis of the significance of differences in HCL in the tested pets depending on the strength of the relationship with the owner was carried out using the Mann–Whitney U test when comparing two groups and the Kruskal–Wallis test when at least three groups were compared. The results were considered significant when *p* ≤ 0.05.

### Methodological Limitations

At the initial stage of planning the research, we assumed that women who had 1 dog and 1 cat at home would take part in the study. As it turned out, however, very often a woman who has both a dog and a cat at home constantly expands her “herd” with new individuals. For this reason, our research group consisted of a total of 25 owners, 55 cats, and 45 dogs. The unequal number of specimens in households made it slightly difficult for us to later perform statistical work on the results. Companion animals are usually treated by the owners as family members and as a separate individuals. The owner has a different kind of emotional relationship with each of the animals, as well as with the children. Therefore, we also treated these animals as separate entities, and did not want to compare the owner’s cortisol level with the average cortisol level of the cats/dogs living on the farm. Therefore, we compared the owner’s cortisol level with each dog separately and with each cat separately.

## 3. Results

Descriptive statistics for human and animal hair cortisol levels (HCL) are presented in Table 1. Spearman’s rank correlation coefficients were determined to investigate the relationship between the caregiver’s and the animal’s cortisol levels. Based on the study conducted, no significant relationship can be found between the HCL of owners and pets (both dogs and cats), as well as between dogs and cats living together (Table 2).

Based on the study conducted, no significant correlation can be found between the HCL of the owner and the animal, broken down by species and sex, or between dogs and cats living together, broken down by gender. Only a significant negative correlation (R = −0.461, *p* = 0.023) was observed between the HCL in owners and male cats.

### 3.1. The Strength of the Correlation between the HCL in the Owner and the Animal Depends on the Frequency of Interactions

The study found no significant correlation between the owner and the dog’s HCL in any of the groups depending on the frequency of kissing the dog. In the case of cats, a significant positive correlation (R = 0.686, *p* = 0.0096) was found between human and cat cortisol levels when cats are never kissed. For the remaining groups, depending on the frequency of kissing cats, no significant correlation could be found between the level of cortisol of the owners and cats.

There was no significant relationship between the HCL in the owner and the dog or cat in any of the groups depending on the frequency of playing with the animal. Likewise, for the frequency of giving your dog/cat treats, the frequency of hugging the animal, and the frequency of having the animal with you while you relax. The study did not show a significant relationship between the HCL in humans and cats, in any of the groups, depending on the frequency of grooming treatments. All analysis results that are not statistically significant can be found in Appendix A. As for dogs, a significant positive correlation was observed between the HCL in the owner and the dog, for dogs in which the owner performs grooming treatments once a week (R = 0.836, *p* = 0.005).

### 3.2. The Strength of the Correlation between the HCL in the Owner and the Animal Depends on the Emotional Relationship

We can find a significant positive correlation between human and cat HCL in the group of people who do not have any special conviction that the cat will be with them even when others leave it (R = 0.576, *p* = 0.031). In the study, there was a tendency towards a negative correlation between human and cat cortisol levels, where the cat helps to survive difficult times (R = −0.335, *p* = 0.095). We observe a tendency for a negative correlation (R = −0.450, *p* = 0.092) in the case of people whom the cat definitely keeps company. There was also a tendency for a negative correlation between human and cat cortisol levels (R = −0.754, *p* = 0.084) in the group of people who want to have their cat constantly nearby.

There is a tendency for a positive correlation between the HCL in the owner and the dog when the dog is definitely next to the human when she needs comfort (R = 0.321, *p* = 0.090). We can find a significant positive correlation (R = 0.526, *p* = 0.036) between the HCL in the owner and the dog, in the case of people who definitely happen to tell their dog what they would not tell anyone else. We also observe a tendency for a positive correlation (R = 0.401, *p* = 0.099) in the case of people whom the dog definitely keeps company, and a tendency for the occurrence of a negative correlation (R = −0.772, *p* = 0.072) in the case of people who do not keep up a companionship with the dog. We can find a significant negative correlation (R = −0.741, *p* = 0.036) between human and dog HCL in the case of people who deny that the dog constantly observes them and focuses its attention on them.

The study showed a significant positive correlation (R = 0.583, *p* = 0.009) between the HCL in the owner and the dog, in the case of people who have a definite feeling that the dog gives them a reason to wake up each morning. However, quite the opposite is the case for cats. There was a significant negative correlation (R = −0.704, *p* = 0.007) between human and cat cortisol levels for people who feel that the cat gives them a reason to wake up each morning.

The study did not show a significant relationship between the HCL in dogs and their owners, depending on how traumatic the dog’s death would be. However, in the case of cats, there is a tendency to observe a negative correlation between the level of cortisol of the cat and its owner when the owner firmly believes that the death of her cat will be a traumatic event for her (R = −0.303, *p* = 0.097).

## 4. Discussion

Cats and dogs are the most popular companion animals. They have fully adapted to the human social environment and are capable of establishing long-term social relationships with people [37]. The influence of the emotional connection on the hormonal interactions between humans and animals is the subject of much research. In our study, no significant relationship was found between the owner and pet’s hair cortisol levels, both for dogs and cats. Similarly, in the studies by Höglin et al. [38] it has not been found that the HCL of the owner is mirrored by the level of this hormone in the dog. However, in previous studies by these authors, a significant correlation was observed between the levels of cortisol in the hair of dogs and owners. However, the dogs participating in that study [6] were shepherd dogs, and the observed interaction, as the authors themselves indicate, may have resulted from the selection of these dogs for cooperation with humans [38]. Mutual understanding is stronger in the human–animal relationship the more time they spend together performing the same tasks. Commitment to training and time spent training together are associated with experiencing a close relationship and this may cause a stronger hormonal dependence [3,39,40,41]. In our study, a significant positive correlation was observed between the HCL of the owner and the dog when dogs were groomed at least once a week, which is also related to spending time together.

In our study, we also did not observe a significant correlation between the levels of cortisol in dogs and cats living in the same house. In addition, we asked the owner if, according to them, the pets live in harmony and if there were any conflicts between the pets. We found no significant correlation depending on the answers to these questions.

The social skills of domestic cats in the context of human–animal interactions have not been studied as thoroughly as for dogs [37]. Perhaps it is related to their shorter period of domestication and living with humans, as well as their higher sense of independence [10]. Even if we consider that cats and dogs have different predispositions to interact with humans, both species are able to communicate effectively with humans in different situations, and perform it differently, because, among other things, humans have developed a completely different type of relationship with these pets [42].

The results of our research suggest that if a human is strongly emotionally connected with a dog, then we observe a different hormonal relationship between them than in the case of a human strongly emotionally connected with a cat. If the owner thinks her dog gives her a reason to wake up each morning, we see a significant positive correlation in their cortisol levels. The opposite is true for humans and cats. If the owner believes that the cat is important enough to her to give her a reason to wake up in the morning, this significant correlation is negative. The more the owner is emotionally connected with her cat (the cat definitely helps her to survive difficult times, wants to have the cat always nearby, and his death will be a highly traumatic event), the more frequent the tendency towards a negative correlation of cortisol levels. Is it possible that the more attention the owner pays to the cat, the more stressful it is for the animal? Might it be that the stronger the emotional relationship between a person and a cat, the more discomfort it causes for the cat?

When the dog is constantly accompanying its owner, there is a tendency for a positive correlation between cortisol levels. Again, the situation is quite the opposite for the cat. If in the opinion of the owner, the cat is still accompanying her, there is a tendency for a negative correlation. Perhaps it is worth considering whether, in this case, a cat following a person and watching him might not be a symptom of passive aggression, as in the case of a cat–cat interaction [43]?

In a study by González-Ramírez and Landero-Hernández [44] comparing the relationships between humans and dogs and humans and cats, the respondents indicated greater emotional closeness with their dogs than with cats (they noticed greater social support, companionship, and unconditional love in their dogs). The relationship of the owners with their cats was assessed as requiring less responsibility and associated with fewer restrictions in daily activities. In a study by Morris et al. [45], cat owners viewed their pets as less emotional and intellectual compared to dog owners. Could this be because humans just cannot read or misinterpret emotions in cats [46]? In our study, owners often felt the same way about a dog as they did about a cat, but it triggered a completely different type of correlation (positive and negative) in cortisol levels.

Most of the research on hormones and stress, both in nonhuman animals and humans, has focused on the sympathetic system and the HPA axis. Although the sympathetic-adrenal system and the HPA axis are widely regarded as the most important physiological systems activated during stress, there are many peptides involved in the stress response in addition to these so-called classical stress systems. These include, but are not limited to, corticotropic releasing hormone (CRH), vasopressin (AVP), adrenocorticotropic hormone, opioid peptides, oxytocin, and several appetite-regulating hormones such as orexin, neuropeptide Y, agouti-related peptide, leptin, and ghrelin [47]. Recently, the effects of the neuropeptides kisspeptin, dynorphin, and neurokinin B, known as KNDy peptides, have also been studied in stress assessment [Ralph et al. 2016]. To fully investigate potential mechanisms underlying the activity or control of the HPA axis, measurements of both HPA direct axis hormones and brain chemicals (e.g., serotonin) that contribute to HPA axis activity would be required [48]. Serotonin is thought to interact complexly with dopamine and cortisol, and in general, serotonin can be said to increase dopamine production and inhibit cortisol production [49]. The role of adrenal steroids involves many interactions with the neurochemical systems in the hippocampus, including GABA in addition to serotonin [50]. Stressful conditions lead to increased free radical production. The increase in free radicals strongly affects all body systems, therefore oxidants and antioxidants arouse wide interest in biological and medical research. Disorders in the pro-oxidative-antioxidant system have been defined as oxidative stress [51]. Although our study did not find many significant correlations between the hair cortisol levels in owners and their pets, studies using other stress markers might have yielded different results.

## 5. Conclusions

Today, there is no doubt that both dogs and cats can create social relations with their owners. They also create a specific emotional relationship with them. The existence of hormonal correlation between humans and pets under acute stress conditions has been repeatedly confirmed in other studies. In our research, no significant correlations were found in the hair cortisol level between humans and dogs, humans and cats, or between dogs and cats living in one household. However, hormonal dependencies regarding chronic stress markers still require in-depth analyses. Broadly understood, human–pet interactions will remain an area of interest for scientists and a wide field of research for a long time to come.

## Figures and Tables

**Table 1 animals-12-01472-t001:** Descriptive statistics for human and animal hair cortisol levels (HCL).

Variable	Mean	Std. Dev.	Minimum	Maximum	Median	Lower Quartile	Upper Quartile
Human HCL (ng/mL)	4.62	1.87	1.88	7.89	4.31	2.99	6.09
Dog HCL (ng/mL)	0.26	0.12	0.12	0.58	0.23	0.19	0.33
Cat HCL (ng/mL)	0.45	0.36	0.18	2.69	0.33	0.26	0.54

**Table 2 animals-12-01472-t002:** Spearman Rank Order Correlations for human and animal hair cortisol levels (HCL).

Pair of Variables	Spearman R	T (N-2)	*p*-Value
Human HCL (ng/mL) and animal HCL (ng/mL)	−0.033	−0.325	0.746
Human HCL (ng/mL) and dog HCL (ng/mL)	0.049	0.319	0.751
Human HCL (ng/mL) and cat HCL (ng/mL)	−0.102	−0.744	0.460
Cat HCL (ng/mL) and dog HCL (ng/mL)	0.115	1.299	0.196

## Data Availability

The data presented in this study are available on request from the corresponding authors.

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
