# Peer review of "Are Hair Cortisol Levels of Humans, Cats, and Dogs from the Same Household Correlated?"

_animals, 2022, doi:10.3390/ani12111472_

Round 1

Reviewer 1 Report

It is ok, but I am still not sure what the authors are correlating in Table 2 "human and animal" since there are 100 animals on 25 humans. If each animal is correlated with its owner (and why then that would contribute to more info than just Human-dog; Human-cat), or if the authors use a "mean animal HCL" for each household if there are more than one animals/household.

Also - I found no answer to my point 16.

Author Response

REVIEWER 1 second response

It is ok, but I am still not sure what the authors are correlating in Table 2 "human and animal" since there are 100 animals on 25 humans. If each animal is correlated with its owner (and why then that would contribute to more info than just Human-dog; Human-cat), or if the authors use a "mean animal HCL" for each household if there are more than one animals/household.

If there was one owner in the household with cortisol X level, we correlated this X cortisol level with every cortisol level of every dog and every cat. In pairs. We did not use the mean pet cortisol level in a given household.

Also - I found no answer to my point 16.

We apologize for omitting the answer to point 16.

  1. In the methods is says that they compared groups but I do not find this in the results.

We have included all the results (including those that are not statistically significant) in supplementary material. There, you can see the division into groups depending on the owner's answers to individual questions. It is these groups that we compared. We think it is clear now.

for example

How often do you kiss your cat?

N

Spearman
R

t(N-2)

p-value

never

13

0,686

3,128

0,0096**

once a month

4

-0,316

-0,471

0,684

once a week

5

-0,158

-0,277

0,800

once every few days

8

-0,422

-1,139

0,298

at least

once a day

25

-0,206

-1,007

0,324

How often do you kiss your dog?

N

Spearman
R

t(N-2)

p-value

never

11

0,035

0,104

0,919

once every few days

7

0,218

0,500

0,638

at least

once a day

27

-0,008

-0,041

0,967

Reviewer 2 Report

The manuscript has been significantly improved and now warrants publication in Animals.

Author Response

We thank the reviewer for his/her comments.